# Superdirective Wideband Array of Circular Monopoles with Loaded Patches for Wireless Communications

**DOI:** 10.3390/s23187851

**Published:** 2023-09-13

**Authors:** Ping Lu, Zhiwei Liu, Enpu Lei, Kama Huang, Chaoyun Song

**Affiliations:** 1School of Electronics and Information Engineering, Sichuan University, Chengdu 610064, China; 2020222055218@stu.edu.cn (Z.L.); 190910438@mail.dhu.edu.cn (E.L.); kmhuang@scu.edu.cn (K.H.); 2Department of Engineering, King’s College London, London WC2R 2LS, UK; chaoyun.song@kcl.ac.uk

**Keywords:** antenna arrays, circular monopole antenna, superdirectivity, wideband

## Abstract

A wideband superdirective array, composed of a two-element circular monopole configuration, is introduced. The monopoles are placed in close proximity, facing each other on a metal ground. To ensure good matching at high frequencies, two pairs of elliptical patches are added to the sides of the monopoles, enhancing the surface current of the circular patch for wideband performance. To achieve equal amplitude excitation and the desired phase difference, a wideband power divider with a phase shifter is designed to feed the antenna array. Simulation and measurement results demonstrate that the proposed wideband antenna array, operating within the frequency range of 2.94–7.93 GHz, exhibits a maximum directivity of 8.36–10 dBi, with an antenna efficiency ranging from 47.86 to 83.18% across the bandwidth. The proposed array has the advantages of miniaturization, high directivity and wideband operation and can be widely used in various portable wireless communication systems, including WLAN (5.05–5.9 GHz), ISM (5.725–5.875 GHz), 5G communication (3.3–3.8 GHz), etc.

## 1. Introduction

Communication performance is limited by fading channel and spectrum resources [1,2,3]. For a limited-space device, a compact size is required [4,5,6]. To overcome these challenges, the current trend in antenna design focuses on achieving high directivity, broadband and miniaturization, which are crucial for meeting the growing demand for wireless communication systems. It was reported in [7] that the normal directivity of an antenna can attain D_n_ = (ka)^2^ + 2ka, where k is the wave number and a is the radius of the smallest sphere that encloses the antenna. Superdirective antenna arrays can be designed to achieve higher levels of directivity compared to conventional antenna arrays, improving communication distance and quality [8,9,10]. Especially for devices in tight spaces, the physical size of the superdirective antenna is minimized while achieving high directivity [11,12].

Although many superdirective antennas/antenna arrays have been reported, a tradeoff can be gained among antenna/array size, gain/directivity and bandwidth. For example, the superdirective antenna/antenna arrays in [13,14,15,16] have the characteristics of small size and high directivity, but these superdirective antenna arrays operate with very narrow bandwidth characteristics, and even a small frequency offset can significantly degrade the performance of the communication system. To solve the problem, several methods have been developed to expand the bandwidth of superdirective antenna arrays. In [17], a magnetic material with a high relative permeability of 99.5 was placed around the dipole antenna to reduce the antenna quality factor, and thus the antenna array bandwidth was expanded. However, the operation of the proposed antenna is limited to low-frequency bands (around 10–20 MHz) instead of microwave bands, and it requires high-permeability magnetic material with a low loss tangent. The use of a non-foster circuit was demonstrated to improve the bandwidth of a superdirective antenna array [18], and the fractional bandwidth (FBW) increased to 80%. The introduction of the non-foster circuit results in high manufacturing costs and a complicated structure, increasing the difficulty of the structure design. To simplify the structure design, a Very High Frequency (VHF) near-field probe based on a Huygens superdirective array was designed. By using a specialized optimization process, the operating frequency band was broadened at the cost of reducing the maximum directivity. However, the operating frequency was still narrow, with an FBW of only 9% [19]. In [20], a ternary superdirective Yagi antenna based on a network feature pattern was designed, and a maximum directivity of up to 8 dBi with 95% antenna efficiency was realized for the frequency below 1 GHz, but its bandwidth was 11.2%. In [21], a butterfly-shaped monopole-based antenna array was reported, and the FBW of the array reached 12.4%. To further improve the bandwidth, an internally loaded superdirective antenna array was proposed [22]. The inductive loadings of the antenna element were optimized for a wide impedance bandwidth (FBW = 32.3%) in accordance with the network characteristic modes theory. Considering the parasitic effect of the lumped element, the proposed antenna array maintains good performance at low-frequency bands. In [23], a wideband superdirective antenna array using plate monopoles was designed for a wide frequency bandwidth (FBW = 83.02%). However, the external power divider with a phase shifter is not discussed.

In this study, a two-element, closely spaced superdirective antenna array loaded with parallel elliptic-shaped patches is designed for wideband operation. The circular monopole provides wideband impedance characteristics at 3.04–5.82 GHz (within the low-frequency band). To further expand the bandwidth, two pairs of parallel elliptic-shaped patches are added to the monopoles. These loaded patches enhance the surface current of the circular patch at high-order modes, enabling resonance at higher-frequency bands. Additionally, a wideband power divider with a phase shifter is designed to achieve the desired excitation. Through careful optimization of the spacing and phase difference between elements, the proposed antenna array achieves a high directivity of 8.36–10 dBi across a wider frequency bandwidth of 2.94–7.93 GHz. Compared to the state-of-the-art designs, our design exhibits miniaturization, high directivity and wideband operation and can be widely used in various portable wireless communication systems, including WLAN (5.05–5.9 GHz), ISM (5.725–5.875 GHz), 5G communication (3.3–3.8 GHz), etc.

## 2. Wideband Superdirective Antenna Array

The superdirective microstrip antenna is arrayed as shown in Figure 1, where the two antenna elements are positioned face to face on the metal ground with a distance of S between them. The circular monopole with a wideband characteristic is chosen as a radiation element and is printed on a TP substrate (high permittivity εr = 24, tan δ = 0.0015, thickness = 0.8 mm). Each antenna element is excited by using a 50 Ω coaxial connector which is connected to the microstrip feedline below the circular patch.

To achieve high directivity over a wide bandwidth, the antenna impedance with different spacing between the elements is investigated, as displayed in Figure 2a, which shows that the simulated impedance bandwidth (|S11| < −10 dB) varies with element spacing S. The bandwidth becomes narrow as S decreases. The influence of element spacing S on antenna directivity is also studied. The results show that the antenna directivity decreases as the spacing increases, as shown in Figure 2b. Although the widest impedance bandwidth (high directivity) can be obtained at S = 28 mm (S = 16 mm), we chose the short distance S = 20 mm for having the highest directivity over broadband [24].

The phase difference between the two elements is studied while the amplitude of the excitations is kept constant over the impedance bandwidth. Figure 2c shows the peak azimuthal directivity with different phase differences (ΔΦ = phase of element #1—phase of element #2) of the array when element spacing S = 20 mm, where the directivity of the single monopole is added. The phase of excitation for element #2 is set at 0°, while the phase of excitation for element #1 is varied between 90° to 180° (for the phase difference ΔΦ < 90°, the radiation pattern tends toward an omnidirectional pattern with low directivity).

It can be seen from Figure 2c that the maximum directivity of the array within the operation bandwidth is relatively stable. Compared to the maximum directivity of a single antenna (5.41–7.26 dBi at the frequency of interest), the array achieves high directivity (D ≥ 8.53 dBi). Over the operating band, an out-of-phase excitation of 150° is chosen to achieve high directivity (9.77–10.19 dBi). The simulated radiation patterns at different frequencies (3 GHz, 4.5 GHz and 6 GHz) are depicted in Figure 2d. The proposed antenna array exhibits a directional pattern with the main beam focused for increased directivity, in contrast to a single monopole antenna that exhibits omnidirectional radiation.

## 3. Wideband Superdirective Antenna Array with Loaded Patches

It is well known that radiation properties and impedance bandwidth can be improved by loading arms or plates [25]. Accordingly, two pairs of elliptic-shaped patches are loaded at two sides of the circular monopole to improve array performance. The wideband superdirective array with loaded patches over an infinite ground plane is dimensioned in Figure 3, where both elements are fed. The circular microstrip antenna typically operates in TM11 mode. The surface current of the monopole with/without loaded patches at different frequencies is analyzed as shown in Figure 4. Without the loaded patches, the current primarily flows at the bottom of the circular patch in the low-frequency bands, while it distributes along the sides of the circular patch in the high-frequency bands. However, with the introduction of the loaded patches, the working mode remains unchanged. The loaded patches have a minimal impact on the array performance at low frequencies due to the small current on the sides of the monopole, and they still contribute to good matching. In the high-frequency range, the surface current of the loaded monopole is enhanced, resulting in improved impedance matching. On the other hand, the weak current of the unloaded patches leads to poor impedance matching [26]. By adjusting the size and position of the loaded patch, an additional resonant frequency is obtained at 8.6 GHz, where high-order mode TM21 is excited, thereby expanding the antenna bandwidth. Additionally, the radiation patterns of high frequency (8.6 GHz) and low frequency (4.5 GHz) are shown in Figure 5, where the simulated maximum directivity of 10.63 dBi is directed at (φ, θ) = (125°, 65°) at 4.5 GHz (low frequency), and 8.01 dBi is directed at (φ, θ) = (136°, 27°) at 8.6 GHz (high frequency). It can be seen that although the radiation pattern at the high frequency (8.6 GHz) is distorted, the main lobe basically maintains end-fire radiation. Using simulation, the performance of the proposed array with different sizes and arrangements of the ellipse patches is investigated as shown in Figure 6. It can be seen that the ellipse patches with optimized size (axis ratio b/a = 1.7) are loaded vertically on the antenna elements, and high directivity and good impedance matching in broadband are fulfilled.

The simulated reflection coefficient |S11| of the two-element monopole array with and without loaded patches for different values of element spacing S is shown in Figure 7a. The array exhibits good impedance matching (|S11| < −10 dB) in the frequency range of 3.23–8.7 GHz, with a fractional bandwidth (FBW) of 91.7%. By introducing the elliptic patches, the bandwidth of the antenna array is significantly improved by 28.9%. Furthermore, the directivity at different S is shown in Figure 7b. It is found that the antenna directivity decreases as the spacing S decreases. Accordingly, the widest impedance bandwidth of the array with high directivity is achieved at an element spacing of 16 mm. The peak azimuthal directivity at different ΔΦ is shown in Figure 7c, where high directivity changes from 8.86 dBi to 10.61 dBi in the frequency of 3–7.5 GHz and from 7.61 dBi to 8.86 dBi in the frequency of 7.5–8.5 GHz at different ΔΦ.

## 4. Antenna Performance and Discussion

To feed the superdirective array, a wideband Wilkinson power divider with a phase shifter printed on the same TP substrate with a thickness of 1 mm is designed for the desired excitation, as shown in Figure 8a. By employing a T-shaped open stub with broadband phase characteristics for the first port, the bandwidth of the configuration is improved, compared to a conventional open stub [27]. For the second port, the transmission line is bent to achieve phase difference ΔΦ. The simulated S parameters of the power divider with a phase shifter at each port are depicted in Figure 8c. The results indicate that the input signal is almost equally transmitted to output ports 2 and 3. A good isolation between ports 2 and 3 is also fulfilled in the frequency of interest. For intuitive presentation, the simulated magnitude imbalance |S21|/|S31| and phase difference ΔΦ (ΔΦ = Φport 3-Φport 2) at 3–9 GHz are displayed in Figure 8d, where port 2 and port 3 output almost the same amplitude signals, and the phase difference ΔΦ between the two ports is 90–180° at 3–9 GHz. Because high directivity is obtained at different ΔΦ over a broadband operation frequency (see Figure 7c), a power divider with a phase shifter is used to feed the superdirective array for the equal amplitude and different ΔΦ. Compared with recently reported SIW feeding networks [28,29], the proposed feeding network (Wilkinson power divider with a phase shifter) can achieve miniaturization with a similar efficiency due to single-layer PCB manufacturing, easy fabrication and low cost.

The proposed superdirective array with loaded patches is fabricated as shown in Figure 9, where the detailed dimensions are listed in Table 1. The reflection coefficient |S11| of the prototype was measured using an Agilent 8527D network analyzer. The simulated (measured) impedance bandwidth of the antenna array with a feeding network is achieved at 2.96–7.99 GHz (2.94–7.93 GHz) (FBW = 91.87% (91.81%)), as shown in Figure 10a. Despite the slight deviation between the simulation and measurement, the two results are in good agreement. The discrepancy of the S parameter between the simulation and measurement may be caused by manufacturer errors. To further understand this discrepancy, the sensitivity of the reflection coefficients |S11| was investigated with different positions h_2_ of the loaded patches, as shown in Figure 10b. The simulated reflection coefficient of the antenna array changes a bit with varying h_2_. Despite the frequency shift, the overall trend remains basically the same. Although a discrepancy in the reflection coefficients exists between the simulation and measurement, the simulation results are in reasonable agreement with the experiments. The radiation performances of the wideband superdirective array are measured in the anechoic chamber. The simulated and measured radiation patterns are shown in Figure 11, where the main beam with the simulated/measured maximum directivity of 10.47 dBi/9.8 dBi is directed at (φ, θ) = (90°, 63°) at 4 GHz (low frequency), 9.6 dBi/9.05 dBi is directed at (φ, θ) = (90°, 68°) at 5.5 GHz (center frequency) and 10.31 dBi/10 dBi is directed at (φ, θ) = (47°, 73°)/(42°, 73°) at 7 GHz (high frequency). The proposed array operates in linear polarization. Since the ground of the antenna array changes from an infinite ground to a square ground with a side length of 400 mm, the maximum directivity of the main beam is directed at a certain angle θ, rather than θ = 90° with infinite ground, as shown in Figure 11.

The antenna array with a feeding network can work at the broadband of 2.94–7.93 GHz (|S11| < −10 dB) while maintaining a good radiation pattern in the frequency band below 7 GHz. It is worth noting that |S11| is the amplitude rather than the phase. With the introduction of a close-spaced array, the phase difference ΔΦ of the feeding network at the two ports changes and is different from that without an antenna array, especially at 7 GHz. To investigate this, we simulated the surface current distribution of the feeding network with and without the monopole array, as displayed in Figure 12. It can be observed that, unlike the current distributions at low-frequency bands, the surface current of the feeding network with the array is significantly different from that without the array at high-frequency bands (7 GHz). The expected excitation provided by the additional feeding network deteriorates with the frequency, resulting in the distorted radiation pattern at the high frequency of 7 GHz. Furthermore, the introduction of a feeding network results in the appearance of a grating lobe, splitting the radiation pattern into two lobes.

The measured maximum directivity of 8.36–10 dBi and antenna efficiency of 47.86–83.18%, which are slightly lower than those in the simulation (directivity: 8.59–10.49 dBi; array efficiency: 47.97–84.72%) over the wide frequency operation, but the tendency of the two results is basically consistent. Regarding the decrease in the antenna efficiency at 7 GHz, it may be caused by the appearing grating lobe with the additional feeding network. The normal directivity as defined by Harrington [7] is compared in Figure 13, which can be calculated with Dn = (ka)^2^ + 2ka, where k is the wave number and a is the radius of the smallest sphere that encloses the antenna. The measured directivity of the antenna array is higher than the Harrington normal directivity limit when the frequency is lower than 4.87 GHz with a bandwidth of 49.42%, indicating that superdirectivity characteristics are obtained by the proposed array.

## 5. Broadband Superdirective Antenna Array Guidelines

The design guidelines of the proposed broadband superdirective antenna array are summarized as follows.

Step 1: Wideband array element design. To achieve broadband, a circular monopole with wideband characteristics is chosen as the array element, as displayed in Figure 1a.

Step 2: Broadband superdirective array design. The two monopole elements are arrayed face to face, as shown in Figure 1b, where both elements are fed. By analyzing the spacing S, the mutual coupling is optimized for the desired bandwidth and directivity. To expand the bandwidth further, two pairs of elliptical radiation patches are loaded on both sides of the radiation patch, as shown in Figure 3. By tuning the loading position and size of the loaded patch, the optimized bandwidth with high directivity is achieved.

Step 3: Wideband feeding network design for the broadband array. A power divider with a phase shifter is designed to feed the broadband superdirective array for the same amplitude and different ΔΦ. Then, the superdirective array with the feeding network is established, as shown in Figure 8b. By optimizing the entire structure, the broadband operation with high directivity is realized.

## 6. Discussion

The performance of this work is fair compared with some recently published wideband superdirective antenna arrays, as listed in Table 2. Compared with other wideband superdirective antenna arrays, our design demonstrates the widest fractional bandwidth (FBW), encompassing the bandwidth of superdirectivity. Despite the presence of isolated resistors on the power divider, the antenna efficiency of the proposed array remains reasonably good. It is observed that the gain and overall efficiency of the proposed superdirective antenna may not be the highest, but the array still exhibits competitive directivity. It is important to note that our design is not focused on achieving a compact array size. Instead, we have taken into account the comprehensive considerations of bandwidth, directivity and efficiency. In the future, superconductivity can be used for these superdirective antenna arrays to enhance antenna gain, and supergain characteristics can be achieved [30,31].

## 7. Conclusions

In this study, we have presented a wideband superdirective monopole array. To achieve the desired excitation, a wideband power divider and phase shifter with a T-shaped open stub is used to feed the broadband array. By incorporating elliptic-shaped patches, we are able to further expand the bandwidth of the antenna array, achieving good matching across the frequency range of 2.94–7.93 GHz (with a fractional bandwidth of 91.8% and 49.42% for superdirectivity) while maintaining high directivity and good antenna efficiency. Overall, the proposed array has the advantages of wide bandwidth, high directivity and good efficiency, and it can be widely used for mobile devices, sensors and small-sized communication devices.

## Figures and Tables

**Figure 1 sensors-23-07851-f001:**
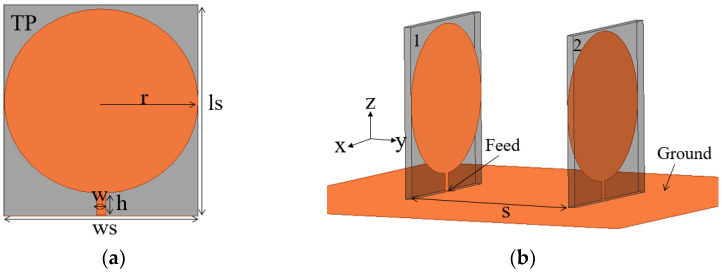
Geometry of superdirective monopole array with both ports being fed; r = 8.5 mm, w = 1 mm, h = 1.9 mm, ls = 20.5 mm, ws = 17 mm. (**a**) Antenna element. (**b**) Superdirective monopole array.

**Figure 2 sensors-23-07851-f002:**
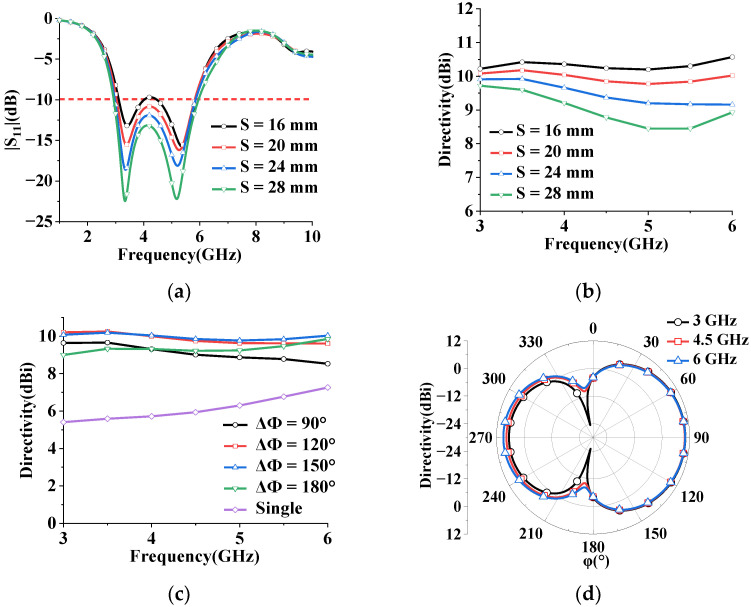
Simulated antenna array performance at different frequencies with both elements being fed. (**a**) Simulated reflection coefficient |S11| with various values of element spacing S. (**b**) Simulated directivity with different element spacing S at ΔΦ = 150°. (**c**) Simulated directivity with different phase differences at S = 20 mm. (**d**) Radiation patterns for θ = 90° at ΔΦ = 150° and S = 20 mm.

**Figure 3 sensors-23-07851-f003:**
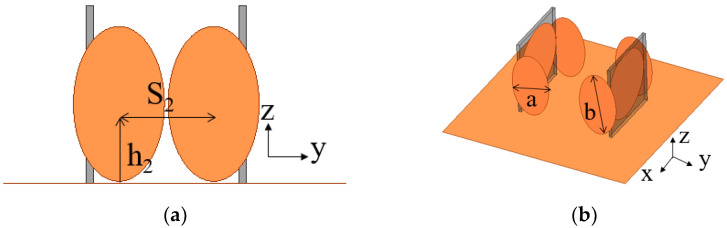
The geometry of the wideband superdirective antenna array with loaded patches. (**a**) YOZ. (**b**) 3D view. S2 = 10.4 mm, h2 = 8.7 mm, r = 9 mm, w = 0.6 mm, h = 1 mm, ls = 19.5 mm, ws = 18 mm. The axis ratio (b/a) of the elliptic patches is 1.7.

**Figure 4 sensors-23-07851-f004:**
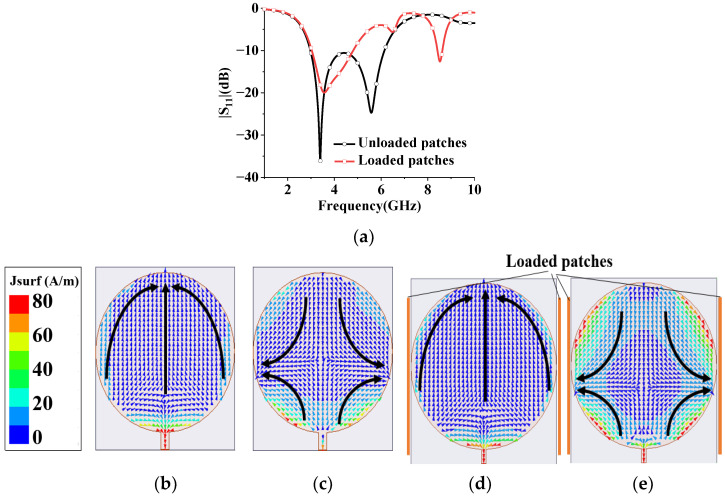
Simulated results of a single monopole with/without loaded patches. The black arrow indicates the current direction. (**a**) Reflection coefficient |S11|. Without loaded patches: (**b**) surface current at 4.5 GHz; (**c**) surface current at 8.6 GHz. With loaded patches: (**d**) surface current at 4.5 GHz; (**e**) surface current at 8.6 GHz.

**Figure 5 sensors-23-07851-f005:**
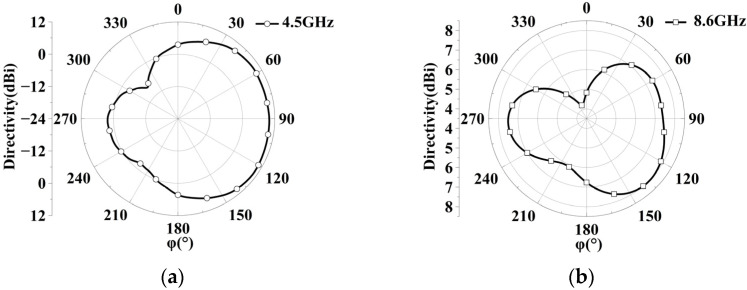
Radiation pattern of the proposed array without feeding network. (**a**) 4.5 GHz at θ = 65°. (**b**) 8.6 GHz at θ = 27°.

**Figure 6 sensors-23-07851-f006:**
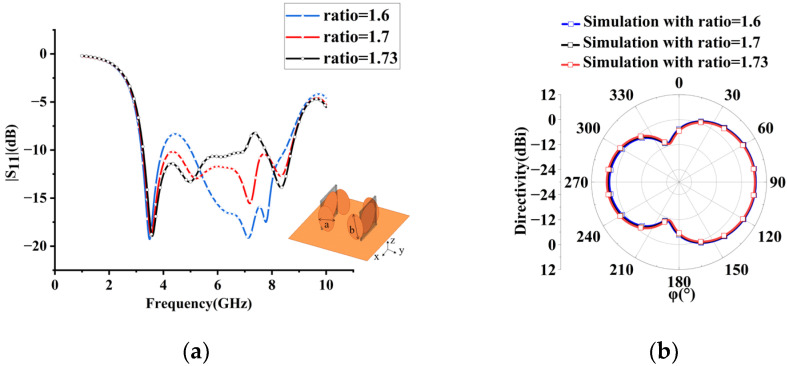
Antenna directivity and S11 for different elliptic patch arrangements (horizontal/vertical) and different sizes. (**a**) S parameter for different elliptic patch sizes. (**b**) Radiation pattern with different elliptic patch sizes at 5.5 GHz for θ = 60°. (**c**) S parameter for different elliptic patch arrangements. (**d**) Radiation pattern with different patch arrangements at 5.5 GHz for θ = 60°.

**Figure 7 sensors-23-07851-f007:**
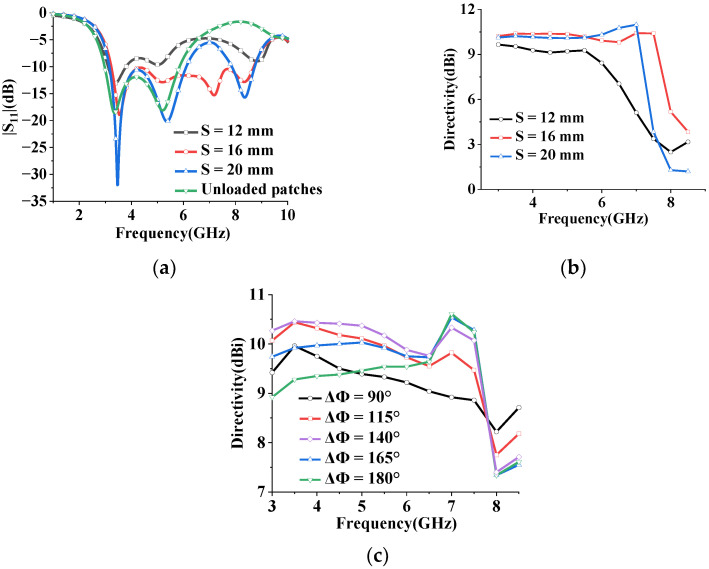
Simulated antenna array performance for different frequencies. (**a**) Simulated reflection coefficient |S11| at different spacing S. Unloaded patch array operates at 3.04–5.82 GHz. (**b**) Simulated directivity with different element spacing S at ΔΦ = 140° for the proposed array with loaded patches. (**c**) Simulated directivity at different phase ΔΦ.

**Figure 8 sensors-23-07851-f008:**
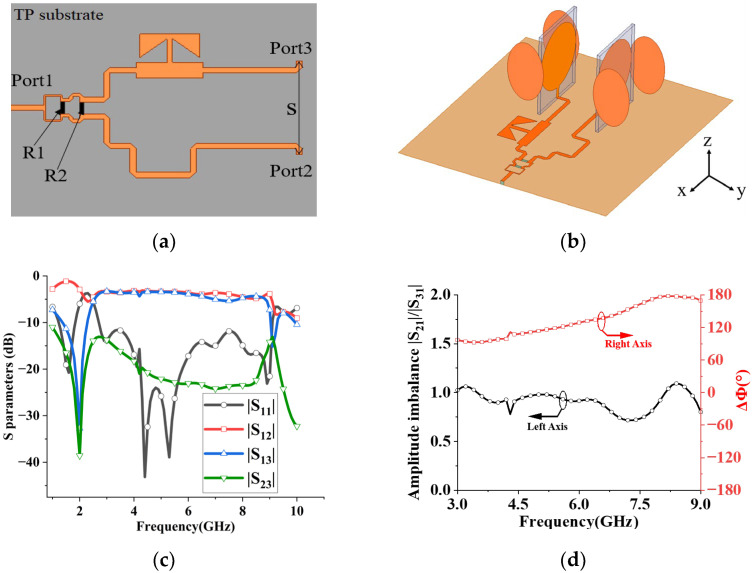
Wilkinson power divider with phase shifter. (**a**) The geometry of the broadband Wilkinson power divider with phase shifter. R1 and R2 are resistors with resistance values of 120 Ω and 240 Ω, respectively, that fix the entire antenna array to the center of the metal ground with a side length of 400 mm. The distance between port 2 and port 3 is S = 16 mm. The power divider with phase shifter is chamfered at each corner. (**b**) Wideband superdirective antenna array with power divider. (**c**) S parameters of the power divider with phase shifter at each port. (**d**) Simulated performance of wideband Wilkinson power divider with phase shifter.

**Figure 9 sensors-23-07851-f009:**
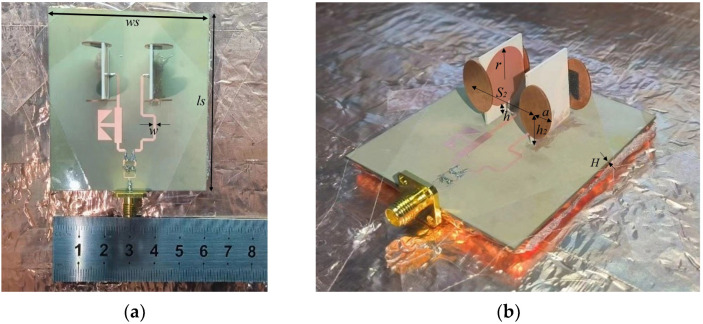
Photographs of the proposed antenna array prototype, where the array is connected with the wideband Wilkinson power divider and phase shifter. (**a**) Top view. (**b**) 3D view.

**Figure 10 sensors-23-07851-f010:**
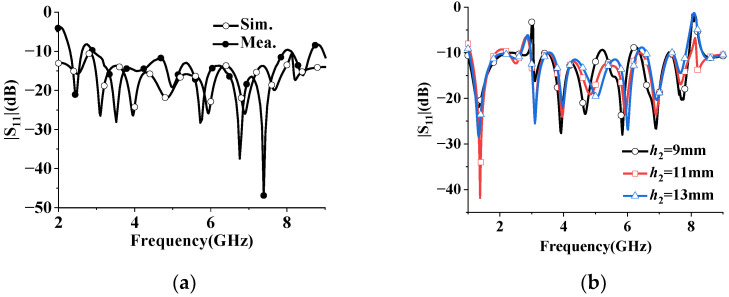
Reflection coefficients of the wideband superdirective microstrip antenna array. (**a**) Simulated and measured |S11|. (**b**) Simulated reflection coefficients at different parameters h_2_.

**Figure 11 sensors-23-07851-f011:**
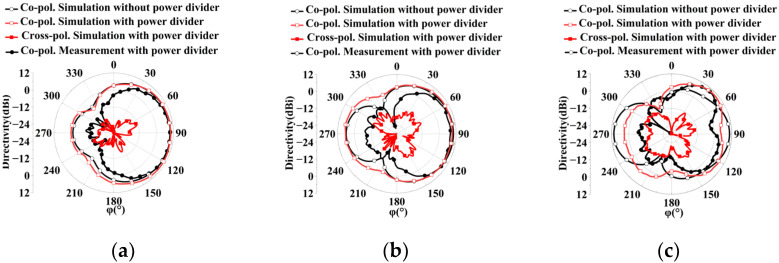
Simulated and measured radiation patterns of the superdirective antenna array at different frequencies. (**a**) 4 GHz at θ = 63°. (**b**) 5.5 GHz at θ = 68°. (**c**) 7 GHz at θ = 73°.

**Figure 12 sensors-23-07851-f012:**
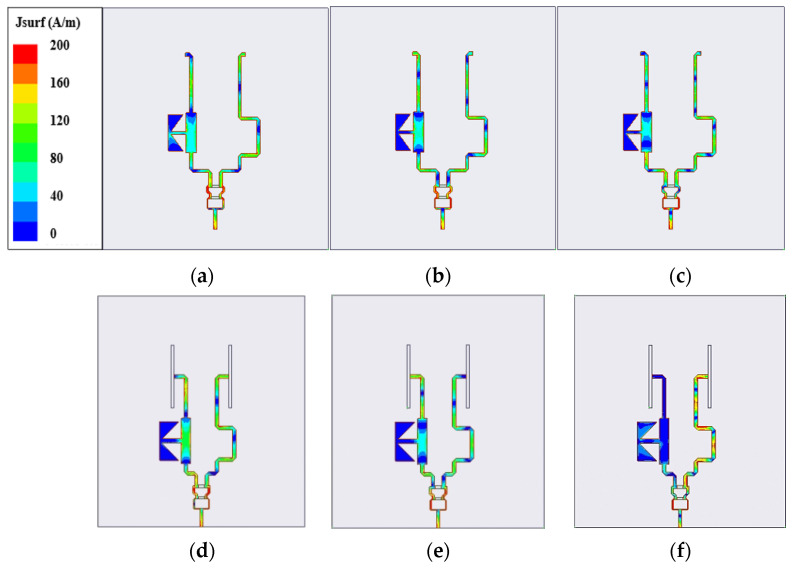
Simulated surface current distribution of feeding network at different frequencies. (**a**,**d**) 4 GHz. (**b**,**e**) 5.5 GHz. (**c**,**f**) 7 GHz. ((**a**–**c**) are only the feeding network. (**d**–**f**) are the feeding network being connected to the antenna array.)

**Figure 13 sensors-23-07851-f013:**
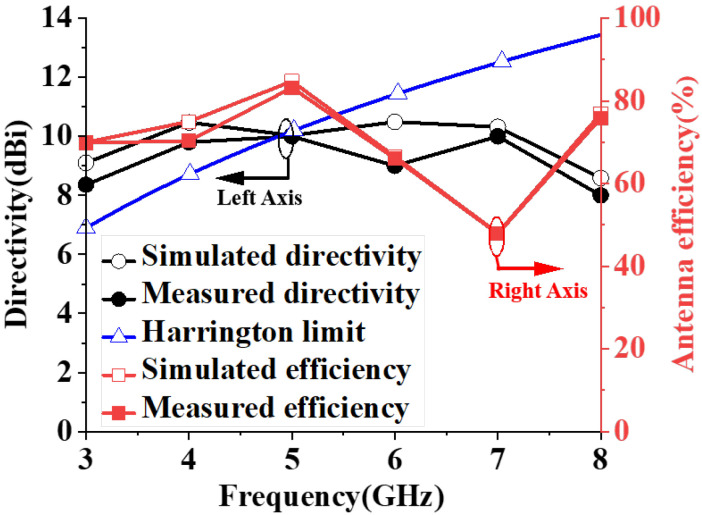
Simulated and measured directivity and efficiency of the wideband superdirective microstrip antenna array.

**Table 1 sensors-23-07851-t001:** Array Parameters.

Structure	Parameters
Circular Patch	r = 9 mm
Microstrip Line	w = 1.2 mm, h = 3.1 mm
Dielectric Substrate	ls = 22.5 mm, ws = 18 mm, H = 0.8 mm
Oval Patch	S2 = 24.8 mm, h_2_ = 11 mm, a = 10.4 mm

**Table 2 sensors-23-07851-t002:** Comparison with Other Studies.

Ref.	Freq. (GHz)	Array Size (Ground Plane Size)	FBW (Superdirective Bandwidth) (%)	Directivity (dBi)	Antenna Efficiency (%)	Antenna Gain (dBi)
[19]	0.154–0.161	0.64λ × 0.64λ × 0.4λ (1.16λ × 1.16λ)	9 (9)	6–12.4	79–89	4.74–11
[20]	0.86–0.97	0.4λ × 0.37λ × 0.18λ (0.47λ × 0.43λ)	11.2 (-)	7.5–8	82–95	6.15–7.6
[21]	0.296–0.335	0.1λ × 0.06λ × 0.21λ (-)	12.4 (12.4)	\	\	8–9.5
[22]	0.78–1.08	0.32λ × 0.34λ × 0.002λ (0.39λ × 0.4λ)	32.3 (-)	−1.5–7	62–85	−1.3–5.6
[23]	3.1–7.5	0.27λ × 0.12λ × 0.22λ (6.2λ × 6.2λ)	83 (25.35)	8.05–8.83	60–91 (Sim.)	4.8–8.0
This work	2.94–7.93	0.34λ × 0.18λ × 0.22λ (3.92λ × 3.92λ)	91.8 (49.42)	8.36–10	47.86–83.18	4.0–8.3

λ refers to the free-space wavelength at the lowest operating frequency.

## Data Availability

Data will be made available on request.

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
