# Peer review of "Superdirective Wideband Array of Circular Monopoles with Loaded Patches for Wireless Communications"

_sensors, 2023, doi:10.3390/s23187851_

Round 1
Reviewer 1 Report
In this paper, a wideband superdirective array, composed of a two‐element circular monopole configuration, is introduced. The monopoles are placed in close proximity, facing each other on a metal ground. Simulation and measurement results demonstrate that the proposed wideband antenna array, operating within the frequency range of 2.96‐7.93 GHz, exhibits a maximum directivity of 8.36‐10 dBi, with an antenna efficiency ranging from 51.29% to 92.04% across the bandwidth.
Overall, the paper is well organized and has good results. Though, I have some comments.
· Can you provide more information about the motivation behind this research and the specific problem it aims to address? What led the authors to choose this particular topic for investigation? It is not evident from the introduction. Novelty is missing. Please revise accordingly.
· What is the significance of this research in the broader context of the field? How does it contribute to advancing knowledge or solving practical problems? Please discuss in detail in the last section.
· Please expand Table 2 with more references.
· What are the key features, frequency range, and performance characteristics of the proposed wideband antenna array, as demonstrated in the simulation and measurement results?
Author Response
We express our sincere thanks for your time and insightful comments on our manuscript.
Our response to your comments can be found in the attachment.
Thank you again.

Reviewer 2 Report
In this paper a wideband directional array, composed of a circular monopole configuration, is presented. Also, a wide band feeding network is designed for equal excitation of two arrays. The topic and concept appears interesting. The idea of wideband and directional array antenna been suggested in many articles published in recent years. Now, it is necessary for the authors to clearly state their design innovations in the revised version. Some major comments are required to be addressed before the acceptance of manuscript:
1) Considering the impedance bandwidth and antenna performance, there is no required to restrict the "for Sub‐6 GHz Communications" in title of paper.
2) The design of the proposed array antenna should be stated in more detail.
3) In this article, has the effect of mutual coupling between the array elements and a solution to reduce its destructive effects in the structure of the proposed array (on the shape of the radiation beam) been investigated?
4) Please compare the size, and efficiency of proposed feed-network with the article "10.1109/TAP.2021.3069557" and "10.1002/mmce.22772" in introduction or result section.
5) Please explain how the reported radiation efficiency of the antenna is measured.
6) The explanation given for the reduction of the radiation efficiency of the array at 7 GHz is not clear.
7) For Table 2, it is recommended to compare the designed antenna array with new articles published in reputable journals.
8) Please explain the Harrington limit curve in figure 11.
9) The radiation polarization of the proposed antenna is not explained.
Author Response

(The authors gave the same response as above.)

Reviewer 3 Report
This research demonstrated a directive antenna array with parallel elliptic‐shaped patches loaded to achieve wideband operation. And loaded patches enhance the surface current of the circular patch at high‐order modes, enabling resonance at higher frequency bands.
Generally, the simulation and experimental discussion were well organized. This reviewer has a few questions for this work.
1. Author added resonance by using parallel elliptic-shaped patches to increase the bandwidth of the super directive antenna. Wouldn’t changing the size of the loaded patches have a significant impact on the bandwidth? It would be nice to show it graphically. Also, is there any reason for arranging the loaded patch vertically at 90 degrees? What is the difference between arranging them on both sides parallel to the main patch, and how big is it?
2. Among the explanations related to the bandwidth enhancement, it is said that the TM21 mode was excited at 8.6 GHz. What about the radiation pattern for this mode (frequency)? I wonder if the super directive radiation pattern is maintained at this frequency as in the low-frequency band.
3. Looking at S11 of broadband Wilkinson power divider and array antenna, it seems to operate well up to about 8 GHz band. However, the explanation that the surface current distribution at 7 GHz is significantly different between the presence and absence of an array and the distortion of the radiation pattern because of this is not well understood. So, does this antenna have a useful frequency band below 7 GHz?
Author Response

(The authors gave the same response as above.)

Round 2
Reviewer 1 Report
It can be accepted.
Reviewer 2 Report
All my concerns have been addressed completely and the revised manuscript paper can be published.
Reviewer 3 Report
This reviewer believes that the revisions and supplements to the reviews have been well reflected in the revised manuscript and that it can be published as is.